# Alternative Food Practices as Pathways to Cope with Climate Distress

**DOI:** 10.3390/ijerph21040488

**Published:** 2024-04-16

**Authors:** Laurence Ammann-Lanthier, Katie Hayes, Iain J. Davidson-Hunt

**Affiliations:** 1Natural Resources Institute, University of Manitoba, Winnipeg, MB R3T 2M6, Canada; iain.davidson-hunt@umanitoba.ca; 2Independent Researcher, Washington, DC 20002, USA; katie.a.hayes@gmail.com

**Keywords:** climate distress, coping, alternative food practices, young adults, climate change, Canada

## Abstract

Experiences of distress and challenging emotions in response to the climate crisis are increasingly common, particularly among young adults. These experiences can include challenging emotions caused by the harmful environmental impacts of conventional food systems, as their contributions to greenhouse gas emissions have become more widely known. While recent studies have examined various experiences of climate distress, the interaction between climate distress and food practice remains poorly understood. In this research, we turn to this intersection by examining the experiences of climate distress of young adults who have alternative food practices, and the interaction between their climate distress and their alternative food practices. Guided by an exploratory, single case study research approach, this research draws from 20 semi-structured interviews conducted with young adults located in urban centres in the Southeastern Prairie Region of Canada. Thematic analysis of the findings reveals that participants experience a variety of climate emotions and a personal responsibility to act in response to the climate crisis. The findings suggest that because of their ability to facilitate a meaningful and practical environmental impact, alternative food practices represent significant climate actions and may be pathways to facilitate coping or managing climate distress among young adults. Results demonstrate the psychological impacts of the climate crisis on young adults, highlighting the need for action on climate change and climate distress. Increasing the accessibility of alternative food practices may support young adults in coping with challenging climate emotions.

## 1. Introduction

Feelings of distress and concern caused by climate change are increasingly common around the world, particularly among young people [1,2,3]. Constituting an emerging area of study, experiences of climate distress are understood as reasonable responses to the climate emergency [3]. One of the concerns of young people experiencing climate distress is the impact of climate change on global food systems wherein climate-related disasters affect ecosystems and habitats, and climate inaction contributes to global greenhouse gas emissions and food waste [4]. Agricultural systems are, at the same time, considered critical to mitigating climate change and regenerating ecosystems [4]. Combined with the ever-increasing urgency for climate action and the need to ensure food security, food systems are considered an important domain of climate action [5]. Drawing from Ammann-Lanthier’s research, in this article we consider the relationship between young adults’ climate distress and their alternative food practices.

### Background

Climate anxiety has recently become an increasingly prevalent concept in pop culture [6], and describes the challenging emotions that arise in response to climate change and environmental issues [7]. Several overlapping concepts have been formulated to understand these emotions, including eco-anxiety, climate trauma, ecological grief, solastalgia, and climate distress [8]. In this research, we use Hayes’s concept of climate distress to encompass a broad range of emotional responses to climate change, including climate anxiety, climate-grief, eco-guilt, climate trauma, and a loss of sense of place due to climate change (solastalgia) [8]. For the purpose of this research, climate distress refers exclusively to non-pathological manifestations of distress, whereby individuals’ abilities to function are not impaired. Non-pathological responses to climate change may include “worry, grief, fear, hopelessness, sadness and general distress related to exposures to climate change hazards and/or knowledge of the climate change problem” [8] (p. 30). As such, climate distress can be experienced solely by the knowledge of the climate crisis, without having the direct experience of a climate change hazard [7,8].

Canada is a country that is warming at twice the global average because of anthropogenic climate change, with extreme climate events such as flooding, forest fires, and drought having already been experienced by Canadians across the country [9]. The impacts of climate change are disproportionally distributed across marginalized communities in the country, as social, economic, political, and environmental inequities impact communities’ exposure and resilience to climate change and climate hazards [10]. Per capita, Canada is the country with the second-largest emissions of greenhouse gas emissions [11], and governments have failed for decades to meet their commitments to reduce these emissions [12] and to adapt their policies to limit global warming to 1.5 °C above pre-industrial levels as per the Paris Agreement [13]. In 2018, direct and indirect household consumption and use of goods and services accounted for 40% of Canada’s greenhouse gas emissions [14], and residential waste accounted for 42% of total waste in 2020 [15]. While actions at the individual level need to be paired with climate policy for effective change towards the 1.5 °C warming limit, these statistics suggest that individual-level pro-environmental behaviours have a role to play in reducing Canadian greenhouse gas emissions and waste. On average, 86.3% of Canadian households engage in more than three pro-environmental behaviours, and in Manitoba, the home province of the majority of our research participants, 71% of households were found to engage in pro-environmental behaviours [16].

The literature on climate anxiety in the Canadian context remains nascent. Surveying Canadians across the country, a recent study found that 73% of respondents believe that we are experiencing a climate emergency, with 50% feeling that climate change is causing mental health issues or worsening these issues [17]. Canadians’ knowledge of and concern for the impacts of climate change is growing [17,18], and the level of concern is likely to increase as the impacts of climate change become more severe. Indeed, Bratu et al. (2022) found that climate anxiety was significantly higher in British Columbia after residents experienced the heat dome created by record-breaking temperatures [19]. In a survey conducted among Canadian youth aged 16–25, Galway and Field (2023) found that respondents reported a diversity of climate emotions, and that 56% felt afraid, sad, anxious, and powerless [20]. Alarmingly, 78% of those surveyed said that climate change impacted their overall mental health. These findings echo those of recent research undertaken in the United States and around the world where younger people (generally encompassing Gen Z between 16–23 and millennials aged 24–39) were found to be more negatively affected emotionally by climate change compared to older age groups [1,2,21,22]. This may be because young people are aware that they will experience the direct impacts of climate change (e.g., future disasters, refugee crises, and food shortages) and face the greatest adaptational challenge [6]. Importantly, research in Canada and around the world finds that the inadequacy of governmental action is one factor contributing to young adults’ experience of climate distress [1,2,17,20,23].

Prevalent in experiences of climate distress is the dissociation from the threat of climate change [3], which can manifest as paralysis or denial [24]. However, scholars have found that non-pathological manifestations of climate distress have the potential of being a productive force, particularly through the exercise of meaningful actions [24,25]. Understanding instances of generative climate distress, or practical climate anxiety, among young people is critical as it can generate an affinity to the incumbent issues and a motivation to contribute to positive change [26].

Drawing from Lazarus and Folkman’s (1984) model of coping strategies, recent studies have examined emotion-focused, problem-focused, and meaning-focused coping within the experience of eco-anxiety in [25]. In emotion-focused coping strategies, the individual addresses their emotional response associated with a stressor, often leading to behaviours that minimize the stressor, such as distancing and avoiding [25]. In the context of coping with climate change, emotion-focused coping does not tend to generate pro-environmental behaviour [8]. In problem-focused coping, the individual responds to a stressor with an action that targets the cause of the stress [25]. Utilizing problem-solving coping to respond to chronic, uncontrollable, and large-scale stressors such as climate change may not be effective because the causes are complex and often out of reach [8,25]. Meaning-focused coping, a third coping strategy later developed by Park and Folkman (1997), is better suited to target such issues in [27], and has been found to be effective in addressing climate distress [8,25]. During meaning-focused coping, an individual explores and renews the meaning of a stressful situation, with “strategies such as positive re-appraisal (acknowledging the stressor but still being able to reverse one’s perspective), finding meaning and benefits in a difficult situation, revising goals, and turning to spiritual beliefs” [25] (p. 2193). Negative emotions are thus acknowledged and confronted, while positive emotions are enhanced [25].

To our knowledge, little has been written about the intersection of food practices as coping strategies to manage climate distress in academic research. Food studies research has identified that participants in alternative food practices prioritize a commitment to sustainable and ethical values [5]. Alternative food practices are activities tied to broader alternative food networks that challenge conventional food systems by their commitments to socio-ecological well-being [5]. Conventional food practices are generally understood as involving conventionally held cooking skills with foods purchased from the existing and often globalized value chains, without consideration of socio-ecological impacts. However, the linkage between commitments to alternative food practices and climate distress has received less attention. Because of the relative accessibility of making changes in one’s food practice, alternative food practices may emerge as pertinent practices for those experiencing climate distress. Authors writing about climate distress have noted that sustainable food choices are one of the actions undertaken by individuals experiencing climate distress. For instance, Zawadzki et al. (2020) found that the consumption of eco-friendly or organic foods contributed to subjective well-being [28]. Furthermore, Ojala and Anniko (2020) found that positive thinking patterns (e.g., believing that one should do the right thing, that one can act as a role model for others, and that one should do the best they can even if others are not taking climate action) were associated with an inclination to make climate-friendly food choices [29]. These works suggest that young adults experiencing climate distress consider food as a valuable arena of climate action. However, the literature is silent on issues of why and how climate distress motivates the adoption of alternative food practices and how such practices impact climate distress, including whether they contribute to self-efficacy, coping, and well-being, issues to which our paper makes an initial contribution.

## 2. Materials and Methods

Ammann-Lanthier’s (2023) research undertook a qualitative single case study of young adults concerned with the environment and who were practitioners of alternative food practices, located in cities in the Southeastern Prairie Region in Canada (see Figure 1) [30]. The conceptual framework developed for this research drew upon climate distress literature in the disciplines of public health and psychology, and the field of food studies. The research was guided by a social constructivist worldview, recognized as appropriate to understand participants’ subjective experiences [31]. A qualitative approach allowed for a focus on participants’ experiences in their own words and generated rich textual description of their experiences, enhanced by undertaking an exploratory case study of one location [32]. A two-tiered sampling approach was utilized by which the criteria were selected first and then the people within the case [33].

The location was chosen based on convenience, as we have connections to organizations in the area that facilitated participant recruitment, and based on the lack of research on experiences of climate distress and/or alternative food practices among urban young adults in the area.

Participants were recruited by emailing local environmental and/or food organizations who were asked to circulate a recruitment poster among their members. Ammann-Lanthier also made a Facebook post on her personal Facebook page that was shared several times. Using purposive sampling, participants were selected if they corresponded to the selection criteria [35]. To be selected, participants had to self-identify as all of the following: a young adult (In this study, participants self-identified as young adults due to the variety of age ranges ascribed to ‘young adult’ across different academic and governmental studies.) living in a city in or around Winnipeg, Manitoba, concerned for the environment, and undertaking an alternative food practice performed routinely and informed by a concern for the environment. The recruitment material used those terms, and also identified that research participants would receive a $50 honorarium for their participation. Snowballing sampling was also utilized, as participants were asked to send a recruitment poster to someone they thought might corresponded to the selection criteria and who would be interested in participating [35]. Due to our familiarity with individuals involved in alternative food practices in the field locations, convenience sampling was a recruitment technique, since we sent the poster to known individuals who confirmed that they met the selection criteria. Regarding the alternative food practices themselves, due to the lack of previous studies examining alternative food practices as climate-informed action, and due to the varying and sometimes contested understanding of sustainability in food practice [5], we chose to let participants self-identify their food practice as ‘alternative’. In the recruitment materials, we provided examples of alternative food practices (specifically: “e.g., growing your own food, canning, baking your own bread, foraging, organizing alternative food procurement, etc.”). By letting participants self-identify, we intended to invite and include a wide variety of practices relevant to our research, and to understand young adults’ subjective experiences in choosing and undertaking these practices as practices informed by their concern for the environment.

Of the 20 participants recruited, 9 self-identified as women, 9 as men, and 2 as non-binary. In total, 11 participants were in the 26–29 age range, 6 participants were in the 30–33 age range, 2 were in the 22–25 age range, and 1 was in the 34–37 age range. As for participants’ ethnic background, 13 self-described as white, Caucasian and/or of European descent, among whom 1 identified Swiss-German roots, 1 identified Franco-Manitoban and Maltese roots, 1 identified Mediterranean roots, 1 identified Mennonite roots, and 1 identified roots in the British Isles. Two participants self-described as Métis, one participant self-described as Filipino-Canadian, one self-described as Latino, one self-described as Colombian, one self-described as having a coloured parent, and one self-described as having African-American, Taino, and Irish roots.

### 2.1. Data Collection and Analysis

Data collection took place from October 2021 to February 2022 and involved 20 one-on-one semi-structured interviews held virtually on Zoom due to the COVID-19 pandemic. In the interview, participants were asked to detail their experiences in alternative food practice and their motivations for undertaking these practices. To understand participants’ climate distress, participants were asked to share how they experience their concern for the environment, including why they are concerned, what emotions they experience as part of their concern, and how their concern for the environment impacts their daily lives. To explore the interaction between their alternative food practices and their climate distress, participants were asked how their concern for the environment influences their alternative food practice, how their alternative food practice impacts their concern for the environment, and what the impact of their alternative food practice is on their well-being (see Appendix A for the interview guide).

The 20 participants generated rich data for the scope and objectives of the study, and this was the point at which saturation was reached. A semi-structured approach ensured participants would address identified themes from the literature, while allowing for flexibility for unanticipated topics [36]. Interviews lasted between 45–120 min, and were audio recorded for transcription and analysis purposes using Zoom’s ‘local recording’ feature. A video recording is automatically downloaded in this function, which was immediately deleted after the interview. Participants chose to remain anonymous or to be identifiable during the process of informed consent, approved by the University of Manitoba’s Research Ethics Committee. If identifiable, they chose the pseudonym or name by which they would be identified. Following the transcription of the interview, the transcript was sent to the participants for review only if they indicated ‘yes’ to the question “would you like to review your interview transcript?” in the informed consent form. In total, 14 participants chose to review their transcript.

Transcripts were then coded and analysed using the NVivo software by Ammann-Lanthier, and the coding structure was reviewed by Hayes and Davidson-Hunt. Data analysis consisted of a thematic analysis approach and an inductive and deductive coding approach. Based on the literature review, deductive and theoretical codes were identified and allowed for a consideration of bias that may have formed during the literature review. In the first round of coding, data was coded for the main themes used to categorize findings (e.g., climate emotions, motivations for alternative food practices), while also creating codes inductively when they emerged [30]. In the second round of coding, inductive codes provided the focus for analysis and emerged from the participants’ experiences, as expressed in the text of the interview transcripts. No disagreements arose among the authors on the coding structure or the analysis.

### 2.2. Ethical Statement

This research was approved by the University of Manitoba’s Research Ethics Committee. Informed consent was obtained before the interviews as participants filled and signed a consent form sent by email, and one participant chose to proceed with the verbal consent form. Consent forms and identifying information were kept in a secure, password-protected document.

## 3. Results

### 3.1. Experiences of Climate Distress

One of the selection criteria for recruitment was that participants self-identified as being concerned for the environment. In the interviews, participants described experiencing an array of emotional responses to the climate crisis, its environmental losses, and the anticipated losses of the future. When asked about how they experienced their concern for the environment, half (*n* = 10) of the participants described their direct relationship to the physical degradation of the environment, such as noticing changes in the seasons, temperatures, and ecosystems of their surroundings, while fourteen participants described experiencing their concern and climate emotions through exposure to climate-related media, documentaries, social media, and scientific reports. In total, 65% of participants (*n* = 13) indicated that part of their concern for the environment was distress caused by governments’ failure to act for planetary health and/or governments’ impediment of progress towards better futures. In total, 15 participants identified a combination of ‘positive’ and ‘negative’ climate emotions, in that they felt grief/anger/worry for environmental losses at the same time as motivation and excitement for the transformative changes required. This motivation was for the opportunities for transformative change through food practice, although some participants also said they were motivated by the transformative change brought by non-food-related climate action (e.g., reducing car use). Participants identified the motivating small-scale changes brought upon by their alternative food practices (e.g., increased plant, animal, and insect biodiversity in their garden, upcycled food waste, or community-building around alternative food practice), or the exciting possibility for broader food systems transformation to which they contributed through their alternative food practices.

Participants understood the need for urgent action in response to climate change to restore ecological well-being (Ecological well-being is understood as the health of humans, species, and natural systems, where “harmonious relationship […] leads to a successful management, distribution, and sustainability of environmental resources for current and future generations.” [37] (p. 1784).), and they felt a responsibility to contribute to this response. As part of this responsibility to act, 13 participants described their care for and/or sense of altruism towards other humans, living beings, and nature. In turn, participants made adjustments to their lifestyles and daily routines to reduce their environmental impact, including adopting alternative food practices, reducing household waste, reusing materials, using active transportation, not travelling by plane, conscious and reduced use of non-renewable and renewable resources (water, energy, oil), and getting involved in social movements.

### 3.2. Alternative Food Practices as Climate Actions

Because the young adults selected as participants were those whose alternative food practices were informed by their concern for the environment, it came as no surprise that a contributor to participants’ climate distress was their understanding of the environmental harm caused by conventional food systems. Participants described the devastating impacts conventional food systems have on species, biodiversity, and soils, their high usage of non-renewable resources, the high carbon emissions of production, transportation, and waste, the destruction of natural spaces for agricultural lands, and their wasteful nature. Participants also chided the disconnection between individuals and their food in conventional food practices. In their eyes, many practitioners of conventional food practices (e.g., purchasing foodstuff at the grocery store) lacked an understanding of and respect for the origin, ecological processes, and human labour involved in growing and distributing food. In describing conventional food practices, Lori explained:

[conventional food practices are] a huge contributor to climate change and it’s a huge contributor to all of these things that induce fear in my heart, [and] that sense of despair. So I can’t, I can’t in good conscience participate in that. I feel that it’s imperative that an alternative path is carved out.

Seeking to change their environmental impact and contribute to socio-ecological solutions to the climate crisis, participants withdrew from conventional food systems and adopted alternative ways to meet their food needs. A central theme that emerged from the research is that participants understood the food domain as a practical, tangible, and meaningful area of climate action. Participants adopted alternative food practices such as growing their own food, preserving foods through canning/fermentation/dehydration, dumpster diving, purchasing discounted foods, getting involved with a local food organization, purchasing locally/organically/regeneratively, and having a vegetarian/vegan diet. The nature of the ‘alternative’ varied per participant, based on their understanding of sustainability and environmental priorities, and based on their personal interests and desires. Altogether, participants’ alternative food practices were characterized by common meanings of sustainability, social responsibility, and food systems transformation in response to climate change and environmental degradation. They involved seeking out materials that have positive ecological and social impacts, and required learning new skills to source, process, and/or produce these alternative foodstuffs. These practices required regularly performed tasks such as baking bread weekly, tending to the garden daily, attentively sourcing foodstuff from ethical and environmentally friendly producers and retailers whenever necessary, processing foodstuff harvested from gardens, dumpsters, or forests, in addition to the daily processes of cooking or preparing these foods. Most practices were undertaken alone or with housemates, such as gardening, baking, and preserving, while dumpster diving and volunteering at the community garden were generally performed in groups (for more details on alternative food practices, please see [30]). Participants understood their alternative food practices as both practices of self-provision and as actions contributing to socio-ecological change. Reckoning with the social injustices entangled in environmental exploitation and degradation, most participants (*n* = 14) communicated their intersectional understanding of environmental issues, whereby environmental actions should be undertaken for environmental and social well-being, and in recognition of how multiple systems of oppression intersect and compound in exacerbating social injustices [38]. Accordingly, participants’ climate distress and subsequent alternative food practices were intrinsically tied to concerns, ethics, ideas, and commitments to act for sustainable and just food futures. For instance, Lex explained how their alternative food practices facilitated a deeper connection to the environment, motivating their engagement in socio-environmental change:

[Feeling] really connected to natural communities of living things, plants, mushrooms, etc. […] really heightened my concern about the environment and my desire to be engaged with the question of what do we do now, and how do we resist the worst impacts of industry and support the people on the ground doing good work that is having positive impacts. And so that has become a more central part of my life; certainly, I think it’s picking up some of these food practices, they drove some of that for me (Lex).

Participants understood alternative food practices as more socially just than conventional food practices. Importantly, they engaged in these practices as a form of meaningful lifestyle activism to bring about positive socio-ecological changes. While we did not specifically ask about participants’ engagement in food-related social justice, this theme emerged in a minority of participants who said they contended with social injustices within their alternative food practices. This involved attempting to make the community garden more welcoming to marginalized people and the produce affordable to low-income populations, or giving away loaves of home-baked bread or tinctures made of foraged plants to low-income people. Likewise, though we did not ask if participants also engaged in collective action or activism for broader food systems change, a minority did add that they also partook in such activities.

In addition to motivations for social change, participants described enjoying and being motivated by the benefits alternative food practices brought to their overall well-being through feelings of joy, peace, and/or satisfaction, and as they enjoyed the meditative and/or physical tasks of their practice (Depending on the practice, the tasks could include watering the garden or walking through the forest looking for mushrooms or berries as examples of tasks described as meditative, or digging garden beds, dumpster diving, and processing a large volume of harvest as examples of tasks described as more physically demanding), the social relations formed with family, friends, and other practitioners with whom they exchanged knowledge and foodstuff, the nutritious food yielded, the ability to live more in alignment with their values, and the increased meaningful connection to nature and food. As such, alternative food practices arose as a pathway for shared benefits to socio-ecological well-being and participants’ personal well-being through their physical, relational, and psychological benefits.

### 3.3. Alternative Food Practices as Potential Pathways to Manage Climate Distress

A major theme that emerged from the interviews is that alternative food practices may lessen participants’ climate distress by allowing them to contribute to environmental health. Participants saw the physical impacts of their practices, for example: changes in the ecosystem, foods withdrawn from landfills, and foods created from scratch with ethical ingredients and without plastic use. In turn, participants (*n* = 11) reported feeling empowered and having more control over their environmental impact through their food skills. For instance, in responding to the question of how her gardening practice impacted her concern for the environment, Evelyn Rose described how she was:

[…] feeling concerned about water, fresh water, then being able to collect rainwater: that, it’s having a response. Having a response to the crisis, and having a response to the concern, so it just doesn’t have to fester. I mean it’s still there, absolutely, but at least it… hmm, what is the feeling? I think it’s empowering, yeah.

Respondents signalled that having self-efficacy by tangibly addressing the problem causing distress eased climate distress. In addition to the positive emotional impacts of self-efficacy, half of the participants said that performing their alternative food practices was therapeutic: the calming tasks and/or the intimate interaction with foodstuff and nature reduced stress and provided relief from climate distress. Furthermore, even as the climate emergency brought into question the viability of the future and one’s responsibility within it, most participants (*n* =14) raised their hope for the future when discussing their climate distress and/or their motivations for prefiguring social change. For example, after identifying fear and despair as climate emotions she experiences, Marika added:

I think my coping mechanism is to feel hopeful about a lot, to feel empowered like I‘m doing something, to focus on all of the things I *can* control in the situation, rather than what I can’t.

For these participants, knowing that there are climate actions to be taken, choosing to enact them, and seeing their impact generated hope for climate futures. These findings suggest that participating in transformative change generates hope that change is possible. Finally, seven respondents also signalled that their climate distress was lessened as they saw others engage in climate actions.

While alternative food practices contributed to well-being, participants also described the challenging or negative impacts of these practices. Most participants (*n* =14) highlighted that the tasks of their practice can be tiring, exhausting, and time-consuming. As their practice involved staying up to date with environmental change to ensure positive environmental impacts, participants (*n* = 11) said that being exposed to knowledge on the climate crisis elicited emotions of despair, being overwhelmed and/or guilt. Moreover, while participants described their sense of empowerment, they also wrestled with feelings of powerlessness. The feeling that one’s efforts were having an insignificant impact on the climate crisis was echoed across 11 participants; for instance, Jérémy Baudet wondered “what’s the point of gardening?” when seeing distressing news such as another incident of shipping containers spilling tons of oil into the sea; and Nate affirmed that his actions have at most “a negligible effect on the environment”. Similarly, a smaller number of participants (*n* = 5) expressed that they felt that their climate actions were ‘not enough’ to address the climate crisis, leaving them with feelings of powerlessness and vulnerability. In turn, questioning the value of their efforts was tiring and disheartening, and contributed to climate distress.

Faced with these mixed impacts on well-being, almost all participants (*n* = 17) spoke about ways in which they ensured their responsibility to act did not cause feelings of being overwhelmed, burnt out, and/or cause them to abandon their practice. Firstly, as participants were aware that the climate crisis is a complex, intersectional, and global problem, they chose perspectives that allowed them to value the actions that were available to them. In total, 11 participants emphasized the value of the actions that are “in their sphere of influence”, however limited, and others (*n* = 11) highlighted the value of small steps towards environmental goals, even if these actions did not have substantial impacts on the climate crisis. Participants also left behind an all-or-nothing perspective on climate action, as 11 participants described how their practice and/or their eco-conscious lifestyles were not perfect (e.g., they occasionally shopped at the grocery store, or they continued to use plastic). For these participants, imperfections and exceptions were considered human and did not detract from the overall satisfaction or relief they felt by contributing to environmental health through their food practices. For example, in describing how he experienced his concern for the environment, one respondent said:

I think a lot of people just don’t even know what to do, where to go, how to even approach anything: basically where I was before, with the observing and the reading and like, how do I get to a point where I can do something? And is that something enough? And I mean, technically, something is better than nothing. So it’s kind of just, start (Tyrell Benton).

Furthermore, some participants (*n* = 6) said that they experienced paralysis in the past because of the enormity of the climate crisis, and that their alternative food practices prevented them from being paralyzed once more. They described how alternative food practices generated feedback of positive emotion that lessened climate distress and prevented paralysis, and that doing something in response to the climate crisis prevented paralysis and quelled climate distress. Finally, six participants explained that they prioritized their happiness to maintain their overall well-being and prevent being consumed and overwhelmed by climate distress.

## 4. Discussion

The findings of this research suggest that alternative food practices contribute to participants’ well-being, echoing findings in the climate distress literature that climate actions contribute to well-being [28,39], particularly when combined with hope [24]. However, the findings also suggest that alternative food practices pose psychosocial challenges, as aspects of these practices increase climate distress. The anxiety, distress, and/or sense of powerlessness that emerge from not being able to do enough echoes feelings of inadequacy found in the literature. Such feelings can be caused by the uncertainty or confusion around what individuals can feasibly contribute to mitigating climate change [40]. In turn, this uncertainty can lead to paralysis, disassociation, or denial [24]. Compounding experiences of inadequacy, our research suggests that participants’ climate distress was aggravated by the lack of appropriate responses by the elected representatives who are perceived as those holding decision-making power. Climate distress caused in part by inadequate governmental (in)action is found in the literature [1,2], including in young peoples’ (aged 16–25) experiences of climate distress in a recent survey conducted across Canada [20]. Likewise, Schwartzberg et al. (2022) found that only 17% of Canadians surveyed think the government is doing a good job in addressing climate change [17].

Motivated by their care for planetary health and their responsibility to do their part, participants adopted multiple climate actions and practices, in line with Bouman et al.’s findings that a personal responsibility to act “may be key in translating abstract worries into concrete and personal climate mitigation behaviours” [41] (p. 8). Accordingly, our findings suggest that participants’ climate distress can be described as ‘practical’ climate anxiety. As put by Pihkala, “the so-called negative emotions have their role in the process of adjusting to the ecological crisis and in trying to build more meaningful futures” [42] (p. 93). Alternative food practices were chosen because they provided skills and meanings relevant to participants’ understanding of the climate crisis, and because they empowered them to have greater agency over their environmental impacts, allowing for greater control of and hope for climate futures.

### Finding Balance: Alternative Food Practices as Pathways to Cope with Climate Distress

Faced with the responsibility to negotiate meaningful climate action, participants chose when they committed to pro-environmental practices, and when they resorted to conventional practices that were more environmentally harmful. What mattered, they highlighted, was to try to negotiate an alternative way of being that aligned with their values, to the best of their ability. This commitment to try is also an “ability to live with ambivalence, so that there is a kind of balance between action and rest”, allowing for well-being, and dissolving feelings of inadequacy [7] (p. 14). Identifying alternative food practices as a meaningful and accessible solution to issues of the climate crisis can be described as a problem-solving approach to coping, “that is, searching for information and making plans about what an individual can do about climate change” [43] (p. 925). Through alternative food practices, and as found in Thompson et al.’s study of adolescents experiencing climate distress, meaningfully contributing to climate efforts locally can generate a sense of self-efficacy for global efforts [40]. In turn, choosing and valuing climate actions where self-efficacy is possible (small steps in one’s sphere of influence) is a helpful approach that can generate hope and increase well-being [39]. Clayton et al. (2017) specify that “[c]onnecting climate impacts to practical solutions encourages action while building emotional resiliency” [3] (p. 17). Thus, choosing climate action within one’s reach may prevent the experiences of being overwhelmed, discouraged, or abandoning the practice that can accompany the perception that only large-scale actions are worth undertaking to address climate change [24]. As participant Lee Hrenchuk said, it is satisfying to “know that you’re having that say in what you think is important”.

Honouring their limits and valuing small steps and actions at the accessible scale are ways participants ascribed meaning and importance to these actions, otherwise perceived as futile in addressing the climate crisis. This reappraisal constitutes a critical reappropriation of agency in the face of the climate crisis: it allowed participants to maintain their connection to the earth, and overcome potential inaction and disassociation caused by climate distress. This process describes characteristics of meaning-focused coping: those of exploring and renewing the meaning of a stressful situation and adjusting one’s goals within it [25,43]. Indeed, as the stressors of the climate crisis were beyond one’s control, coping potentially occurred as participants reframed their view of the climate crisis and their role within it in such a way that allowed them to honour their responsibility to act for the well-being of the planet, while also maintaining their own well-being and without dismissing the complex nature of the climate crisis. The prioritization and balance maintained between personal and socio-ecological well-being and action-taking illustrate findings in the literature that climate actions have the potential to contribute to both ecological health and mental health [1,20,28].

In their pilot study on the impact of ambivalence among emerging adults’ sustainable food choices, Ojala and Anniko (2020) find that a positive thinking pattern, having a “focus on the ‘right’ thing to do” rather than on the consequences or limited impacts of climate action, is associated with a higher inclination to undertake sustainable food choices [29] (p. 28). Meanwhile, having higher ambivalence about food choices and negative thinking patterns, including binary thinking, are associated with being less inclined to make sustainable food choices. Similarly, participants in our research showed characteristics of positive thinking patterns: an ability for dialectic thinking (e.g., the ability to choose and value accessible actions while holding mixed beliefs on their impact) and a focus on the ethical character of alternative food practices in their commitment to exercise their responsibility to act for ecological well-being.

Interestingly, Wray highlights that some researchers refer to climate anxiety as a form of climate compassion or climate empathy because it is founded on the care for other beings and because it emerges as “an antidote to the culture of uncare” [6] (p. 53). Referencing psychoanalyst Sally Weintrobe’s analysis, she explains that social welfare has been replaced by a culture that “promulgates the social acceptance of selfish impulses and short-sightedness” [6] (p. 48). In this culture, people “disconnect from parts of themselves that take responsibility in life” [6] (p. 49). Contrastingly, alternative food practices were characterized by abundant and joyful sharing, and much like the concept of climate compassion, by affects of care for nature and living beings that foster a sense of responsibility and solidarity towards these, challenging the ‘culture of uncare’ and norms of self-interested individualism. Indeed, individuals who care for others and nature, and who wish to contribute to society, tend to be more impacted by climate distress [1], while caring for the environment motivates climate mitigation behaviours [41]. More than solely seeking to reduce their quantitative impact on the environment, participants cultivated meaningful and healing human-to-nature connections and relationships through their alternative food practices.

## 5. Conclusions

The participants in this research experienced a range of emotions as responses to the overwhelming complexity and gravity of the climate crisis and its intersectional impacts, the ever-so-urgent need for radical transformation as communicated by scientists and activists, the uncertainty of the future, and the inadequate governmental and societal response. Potentially experienced as practical anxiety, climate distress informed participants’ responsibility to act for environmental health, as they carefully evaluated their life practices to ensure these were as ethical and environmentally friendly as possible. Alternative food practices were chosen strategically because participants felt empowered by the food skills and the subsequent ability to have self-efficacy in shaping their socio-ecological impact and in responding to the climate crisis. The findings of this research suggest that climate distress was lessened through the ability to have more control over one’s ecological impact and through multiple contributions to a sense of well-being. However, participants also identified ways in which alternative food practices strained their well-being. In turn, valuing actions that are accessible to them, however small or imperfect, and prioritizing their personal well-being were ways participants framed their responsibility to act to ensure that their alternative food practices contributed to feelings of efficacy and empowerment, quelling climate distress. As such, for the young adults interviewed, part of adapting to the psychological impacts of the climate crisis was to identify and undertake meaningful practices within the food domain.

While our findings suggest that there are psychological benefits to individual action, we echo other scholars in the climate distress literature who emphasize the need for collective and systemic action on the part of governments to adequately respond to the climate crisis [2,23]. Particularly as climate distress is aggravated or partly caused by governmental failure to act, appropriate and timely action by our elected representative is needed, in addition to individual behavioural changes. Within the scope of this paper, we drew attention to alternative food practices in order to better understand this individual-level behaviour and its potential role as a form of coping with climate distress.

### Study Limitations and Further Research

The findings of this research are exploratory and case-bound, based on a case study of 20 young adults and as such, the findings should not be generalized. While the coding structure was reviewed and approved by Hayes and Davidson-Hunt, only Ammann-Lanthier conducted the coding and interacted with the raw data. Further, the research participants did not review or provide their thoughts on the coding structure and the research findings. While it was beyond the scope of this study to do so, future research could include a focus group with the same research participants to discuss the themes that emerged from the one-on-one interviews.

Our study emerged in response to the increasingly common experiences of climate distress among young adults, and in response to the urgent need to understand the psychological impacts of climate change and identify potential coping strategies. Further study is needed to understand the interaction between food practice and climate distress, and to explore how and why engaging in alternative food practices may constitute a coping strategy. Opportunities for future research on alternative food practices include examining the role of self-efficacy, increased connection to nature, and social networks of support in facilitating psychological resilience to climate distress. This could include comparing the efficacy of different alternative food practices as coping strategies; for example, to assess whether practices performed mostly alone differ from practices performed in groups regarding their impact on coping and psychological resilience. Moreover, as our topic represents an understudied area of the literature, further research should explore how gender, socio-economic status, ethnic background, and other demographic factors shape young adults’ experience of alternative food practices as climate distress-mitigating practices. Our study focused on urban areas of the Southeastern Prairie Region (Manitoba and Northwestern Ontario), which are relatively more isolated from the extreme climate change hazards occurring elsewhere in the country, and around the world. Future research could explore the experiences of climate distress of both rural and urban-based young adults in areas more vulnerable to climate hazards, to explore the relevance of alternative food practices as climate actions and coping strategies to climate distress. As future research examines and expands our understanding of food practices as potential coping strategies, our findings may be relevant to organizations or initiatives seeking to address climate distress among young adults. If alternative food practices are indeed pathways through which young adults can better cope with climate distress, increasing their accessibility or creating opportunities for learning and participation in these practices may be practical ways to support young adults in coping with challenging emotions.

## Figures and Tables

**Figure 1 ijerph-21-00488-f001:**
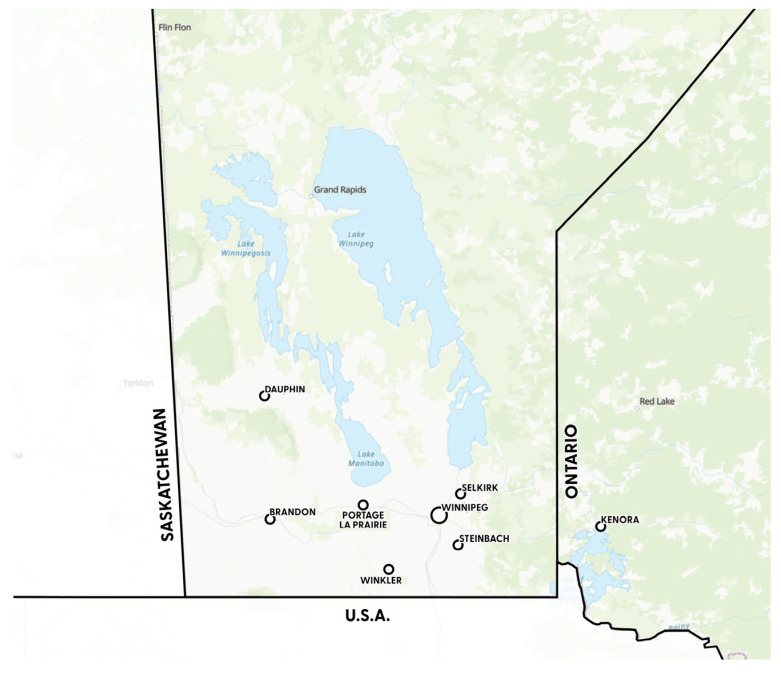
Southeastern Prairie Region [34].

## Data Availability

Coding tables and more information on data collection and analysis can be found in Ammann-Lanthier’s research archived online by the University of Manitoba [https://mspace.lib.umanitoba.ca/items/786059f1-519f-40dc-8c21-8e76ac350d24, accessed on 29 March 2024]. The raw data presented in this article is not readily available because of privacy reasons, as participants did not provide consent to share interview transcripts. Requests to access the datasets should be directed to ammannll@myumanitoba.ca.

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
