# Peer review of "Alternative Food Practices as Pathways to Cope with Climate Distress"

_ijerph, 2024, doi:10.3390/ijerph21040488_

Round 1
Reviewer 1 Report
Comments and Suggestions for Authors
The manuscript presents a timely exploration of the psychosocial dimensions intertwined with environmental issues, making it a suitable candidate for the journal's scope. I have a few comments.
1. While the research's Canadian setting provides a unique opportunity to explore culturally specific responses to climate change, the paper lacks extensive discussion of Canadian examples. The authors should incorporate more instances that reflect Canada's particular cultural context regarding climate change awareness, anxiety manifestations, and correlated environmental behaviors among Canadians.
2. The authors investigated food practices as coping mechanisms in the face of climate-related stress, but their treatment of this core theme is ill-defined. The literature review features only a cursory paragraph referencing three studies. For this focus to be genuinely pursued, an expanded literature review is necessitated—one covering a comprehensive definition of food practices and their pertinent importance as coping strategies during climate crises and associated distress periods. Such literature should also include relevant rationales and theoretical frameworks.
3. Considering the small sample size of 20 participants, the authors must provide a detailed demographic breakdown—gender, age, socioeconomic status (SES), ethnicity, level of education, etc. Given the wealth of research highlighting social justice concerns and inequalities concerning climate change adaptations, complete participant characterization will help readers understand the distinctive case presented here.
4. Regarding qualitative analysis transparency, who performed the coding? Was there any disagreement among coders regarding emergent themes? Disclose such information to affirm methodological robustness and analytical validity.
5. In line with best practices for qualitative work, an interview illustration should include references to participants' gender, age, and ethnicity/race—factors that could profoundly influence personal narratives around climate distress.
6. If present, a sub-sample that did not report experiencing climate-related distress warrants acknowledgment within the findings for contrast and depth. Their perspectives might provide valuable counterpoints aiding in understanding resilience or denial aspects in relation to eco-anxiety.
7. The authors should outline the limitations — possibly pertaining to sample size or representativeness — thus framing their results within appropriate bounds of generalizability.
Reviewer 2 Report
Comments and Suggestions for Authors
I would like to congratulate the authors for an excellent choice of topic as climate anxiety concerns the young generation all over the planet. I find the choice of combining climate distress with alternative food practices original and very interesting because it offers an innovative and optimistic look at the climate crisis research. I find the article well written and understood by a wide audience. I appreciate that you have chosen qualitative methodology for your research. The structure of the research presentation is excellent and detailed. Some suggestions for improvement in individual points are listed.
Initially, I think that in the theoretical part where the climate anxiety of the new generation is presented, perhaps the reference to a key research on this issue is missing:
Hickman, C., Marks, E., Pihkala, P., Clayton, S., Lewandowski, R. E., Mayall, E. E., Wray, B., Mellor, C., and van Susteren, L. (2021). Climate anxiety in children and young people and their beliefs about government responses to climate change: a global survey. The Lancet Planetary Health, 5 (12), 863-873.
Regarding the research participants, it would be interesting if you could provide us with some additional demographic and qualitative information about them if collected such as age, gender and engagement with environmental issues.
It would also be helpful to include key questions that you asked the participants to make the content of their answers easier to understand.
Furthermore, I would like to ask you if there was even one of the participants who raised any concerns in relation to the need for wider political, economic and social changes that should be made in the matter of nutrition (and not only in personal, individual food choices).
Finally, it would be interesting to mention open questions for future research. For example, scholars ( eg. eg Aruta, JJBR. (2022). Letter to the Editor: Mental health efforts should pay attention to children and young people in climate-vulnerable countries. Child Adolesc Ment Health // Tsevreni, I., Proutsos, N., Tsevreni, M., & Tigkas, D. (2023). Generation Z Worries, Suffers and Acts against Climate Crisis—The Potential of Sensing Children’s and Young People’s Eco-Anxiety. Climate) report that there is a deficit in research data from the Global South. Do you think it would be interesting to conduct a comparative study in poorer countries of either the Global South or in more economically and technologically vulnerable areas of Global North to reveal if the new generation there has the same approach or are there small qualitative differences?
Reviewer 3 Report
Comments and Suggestions for Authors
The authors state, “Ojala and Anniko’s (2020) noted that positive thinking patterns were associated with an inclination to make climate-friendly food choices.” What were these positive thinking patterns?
The authors report that they are familiar with those who engage in alternative food practices; however, a landscape of these practices is not provided in the introduction. Is collecting rainwater an alternative practice?
In describing the selection criteria, the authors report, “To be selected, participants had to self-identify as all the following: a young adult living in a city in or around Winnipeg, Manitoba, concerned for the environment, and undertaking an alternative food practice performed routinely and in formed by a concern for the environment.” Please provide demographic information about the participants. Were there any indigenous participants? How much gender and economic diversity was there in the sample? The term “alternative food practices” seems very vague? Was this the term used in the recruitment materials?
Were participants compensated for the interviews? How many coders were involved in the study? How were discrepancies in coding resolved? The pseudonyms are confusing and should be removed. Provide the interview guide as an appendix.
The authors state, “15 participants identified a combination of ‘positive’ and ‘negative’ climate emotions, in that they feel grief/anger/worry for environmental losses at the same time as motivation and excitement for the transformative changes required.” Was this transformational change regarding agricultural and food consumption patterns? Were they doing anything to promote these changes?
Participants understood the need for urgent change towards ecological well-being and felt a responsibility to contribute to this change. What was meant by “ecological well-being?”
Although the focus of the interviews was purported alternative food practices, very little was said about what these practices included. There is one sentence on this subject: “Participants adopted alternative food practices such as growing their own food, preserving foods through canning/fermentation/ dehydration, dumpster diving, getting involved with a local food organization, purchasing locally/ organically/regeneratively, and having a vegetarian/vegan diet.” How much time was spent on these activities? Were they done with others? Were they conducted to address climate change? Were they conducted before there was a major concern about climate change in the media and/or extreme weather events attributed to climate change?
What were examples of meditative and physical practices?
What were the “food communities?”
How much were the respondents involved in collective action to address climate change? Were most of the activities idiosyncratic or part of an organization?
It is well recognized that individual-level behaviors will not have a substantial impact on climate change unless these behaviors facilitate and support collective action and massive policy changes to address climate change.
Was there any discussion of social justice in the food practices?
Comments on the Quality of English LanguageThe article needs editing for spelling and grammar.
Round 2
Reviewer 1 Report
Comments and Suggestions for Authors
The authors have addressed my concerns.
Author Response
We thank the reviewer.
Reviewer 3 Report
Comments and Suggestions for Authors
Please clarify that the use of actual participant names was approved by the IRB. Also, were participants explicitly asked if they wanted to review the transcripts? If so, did any ask?
Author Response
We thank the reviewer.
We have added clarifications at lines 221-222, and 225-226. In Section 2.2 Ethical Statement (line 238-239), we also state that "This research was approved by the [First author’s institutional affiliation]’s Research Ethics Committee", which includes approval of the informed consent form.